# Casualties During Marathon Events and Implications for Medical Support

**DOI:** 10.3390/healthcare13172249

**Published:** 2025-09-08

**Authors:** Juliana Poh, Venkataraman Anantharaman

**Affiliations:** Department of Emergency Medicine, Emergency Medicine Academic Clinical Program, SingHealth Duke-NUS Academic Medical Centre, Singapore General Hospital, Outram Road, Singapore 169608, Singapore; poh.juliana@gmail.com

**Keywords:** heat injuries, sudden cardiac arrest during sports, heat index, incidence rate ratio, strenuous physical activity

## Abstract

Introduction: Marathon runs conducted in tropical environments can result in high injury rates. This study was conducted to provide information about the burden of injuries in such environments, to aid planning for similar mass events, enhance medical support, and improve participant safety. Methods: This was a retrospective review of casualty data from the Singapore Marathon races from 2013 to 2016. Patient Presentation Rate (PPR) and Transport to Hospital Rate (THR) were calculated and correlated with heat index, derived from weather information. Injury types were also reviewed. The negative binomial regression was performed to investigate impact of heat index on casualty rates. The medical response plan is briefly described. Results: During the four-year period covered, heat index increased from 29° to 35°. There were more casualties amongst the participants from the full marathon than other race categories. The THR was 0.3 to 0.68 per 1000 participants. Two participants had cardiac arrest. Negative binomial regression showed significant impact of heat index on casualty rate. Incidence rate ratio was 1.22 for severe casualties, which indicated that every 1 unit increase in heat index resulted in 22% rise in severe casualty numbers. Compared with 10 km racers, half marathon racers experienced 1.58 times greater likelihood of all injuries and full marathon racers, a 3.87 times greater risk. Conclusions: Adverse weather conditions with high-heat index can increase injury rates during strenuous physical activities such as the marathon. Applying careful measures to minimise the impact of heat and high humidity may help minimise such injuries.

## 1. Introduction

Marathon runs as competitive events are held around the world annually. World Athletics (formerly referred to as the International Association of Athletics Federations or IAAF) currently recognises 14 Platinum-labelled Marathon Races, 20 Gold-labelled Marathon Races and 15 Elite-label Marathon races [1]. The Platinum races are all held in temperate areas of the world. Amongst the Gold races, the Singapore Marathon is the closest to the equator at latitude 1.3521° N. The annual Singapore Marathon (SM) started off in 1982 and is always held in the first weekend of December [2]. Besides the full marathon (42.195 km) category, there is also a half marathon (21.1 km) category, relay category (team of 5 participants to cover 42.195 km), a 10 km run, and a 10 km wheelchair category. The Kids Dash (600 m) and 5 km categories were added subsequently. This running event is immensely popular and attracts about 50,000 local and international participants each year. Temperatures in Singapore during the month of December usually range from 25 °C to 32 °C, though maximum temperatures of up to 35 °C have been documented [3]. The average relative humidity in December is also high, usually > 80%. Together, this results in a high-heat index here. Medical support at these marathon running events is often dependent on race organisers and budgets. The medical support plan, if there is one, can vary from onsite first-aiders and tapping on private ambulance providers to involving medical teams with advanced life support capabilities and evacuation to public hospitals. Several participant characteristics can influence casualty presentation rates and medical utilisation during marathon runs, including age, sex, pre-existing health conditions, and training habits. In 2013, the IAAF produced its first edition of medical support guidelines for such events [4]. These provided general recommendations to be used and to be appropriately adapted by race organisers for their individual events.

There have been a number of publications on injuries sustained during marathon or ultramarathon events [5,6,7,8]. These marathon events had all occurred in temperate climates. Very little has been published about injuries in tropical marathons [9,10]. The special considerations for marathon events in tropical environments with heat and high relative humidity may differ from those in the other marathons organised around the world. The objective of this study was to determine the types of injuries occurring in marathon events being conducted in a tropical environment situated very close to the equator and with high temperatures and high relative humidity, so as to provide a basis for medical planning for future such events in similar environments.

## 2. Medical Support Provided from 2013 to 2016

The Singapore General Hospital was contracted to provide medical support at the Singapore Marathon for a four-year period from 2013 to 2016. The marathon routes were largely unchanged over the four years. The event was conducted in the southern and eastern part of the island and ended at the same finishing point. The start times and numbers of participants for the various runs over the four years are shown in Table 1. Command and control of the medical response was provided by a Medical Command Centre with decentralised control over five sectors by senior physicians at each of these sectors. Each sector had a medical team of doctors and nurses at strategically located medical tents along the marathon route with the distances between the tents covered by roving first aid teams consisting of nurses and first-aiders on foot and on bicycles. Each medical tent had full resuscitation capability, manned by doctors, nurses, physiotherapists, and operations support staff. Each sector was also allocated a number of ambulances to facilitate movement of injured participants from locations of incidents to the medical tents and also to hospitals in the vicinity, if required. Approximately 35 ambulances, each with two paramedics and one driver, were on standby at the medical tents and along the running routes, strategically positioned to achieve short response times. There was one ambulance coordinator at each medical tent. The largest medical tent was located just after the finishing point. The medical plan was refined yearly, based on the experience gained from the preceding years and adjusted according to the changes in the routes.

For the first 33 km, two nurses and six to eight first-aiders covered each kilometre. From 33 km to 41 km, there were four nurses and 10 to 14 first-aiders for every kilometre. From 41 km to 42 km, there were four nurses and 20 first-aiders. Along the last 200 m, there were two nurses and 20 first-aiders spread out at 50 m intervals. The nurses were equipped with bicycles for patrols, mobile responder bags, and Automated External Defibrillators (AEDs). There were four nurses and 20 first-aiders at the finishing point.

Upon sighting any casualty along the route, first-aiders would respond and call the respective nurse if additional assistance was required. The call would be escalated to the medical tent doctor if consultation was required. The medical tent doctor would activate the nearest ambulance for evacuation to the medical tent or the nearest hospital, if necessary. Operations staff would record the incident and timings in the Incident Record Log and inform the Medical Command Centre of all evacuations. All tents and key position holders communicated by walkie-talkies on various pre-assigned channels.

Upon registering for the marathon, registrants were given a thick instruction booklet, which included the Physical Activity Readiness Questionnaire. There was no mandatory requirement to complete the questionnaire or submit it to the race organisers. Water points were located at approximately 2.0 km to 2.5 km intervals along the marathon route [11].

The start timings of the various marathon events are as given in Table 1.

## 3. Methods

This was a retrospective study of all casualties sustained during the SMs conducted from the years 2013 to 2016. Data collection was through the use of casualty cards available to all medical and first-aid teams employed during the four events. All casualty cards were eventually handed to the medical teams covering each sector of the marathon event and these were finally passed to the Medical Command Centre. Data collected included the marathon participant number, race category, name, gender, medical tent A/B/C/D/E, time of presentation, presenting complaint, treatment rendered, disposition, and disposition time.

Injured participants were classified into light, moderate, and severe categories as suggested in Table 2. Data from all medical tents and first-aiders was collated into a single casualty record at the end of the event. All casualty identifiers were removed for the purposes of this study. The patient presentations were also correlated with heat index, derived from temperature and humidity.

Data was collected on the race days from 2013 to 2016. The full marathon, half marathon and 10 km race casualties were included in the analysis. The kids’ race and wheelchair race were excluded. In 2013, only data about patients who were managed at the medical tents was collected because of the expectation that most injured participants would be brought to the medical tents for further care and subsequent disposition. This was extended to those seen in the open areas by physiotherapists at the large field near the finishing point for 2014 and 2015 because of the relatively large numbers of light injuries and resources expended to attend to them during the 2013 marathon, though data on these lightly injured participants were not initially collected in 2013. In 2016, data of participants attended to by first-aiders all along the route, even if not referred to the medical tents or not evacuated directly to hospital, was also collected to obtain a more complete picture of injuries sustained during the event.

Weather (temperature and humidity) data was obtained from the National Environmental Agency of Singapore, for the race days, from 5 am to 12 noon, from two weather stations which were along the race route. Weather data was reported as means. Heat index was derived from an online heat index temperature calculator (calculator.net). Only a relatively small portion of the full marathon (about 10 km) occurred in the eastern part of the island. The other races were held in the western part of the whole area used by the marathon event.

The heat index, also known as the apparent temperature, is a measure of how hot it feels to the human body when relative humidity is combined with the actual air temperature. It essentially reflects the human body’s perception of heat, accounting for the cooling effect of evaporation being reduced by higher humidity.

This study received approval from the SingHealth Centralised Institutional Review Board with waiver of requirement for informed consent from marathon participants (CIRB Ref 2017/2864 dated 4 October 2017).

## 4. Data Analysis

The patient presentation rate (PPR) was calculated based on the number of injured participants divided by the number of total participants multiplied by 1000. The transport to hospital rate (THR) was calculated by dividing the number of injured participants sent to the hospitals by ambulances by number of total participants multiplied by 1000.

The basic demographic information of casualties was summarised across the event years and compared by type of competitions using one-way ANOVA for continuous variables and chi-square tests for categorical variables. We initially performed Poisson regression model which showed extreme overdispersion (dispersion ratio = 202.0). Therefore, the negative binomial model was applied to investigate the impact of heat index, adjusted by types of competition, on casualty rates. Model fit was superior for the negative binomial (AIC = 161.7; BIC = 164.1) compared with Poisson (AIC = 1708.6; BIC = 1710.5). For this model statistical software, R, Version 4.3.0, was used in estimating the Incidence Rate ratios (IRR) for the heat index and also for the half marathon and full marathon, using the 10 km run as base. The IRR quantifies how much more or less likely an event is to occur in one group compared to another, based on their respective incidence rates. In the case of a marathon event, using a non-run event was considered inappropriate for calculation of injury incidence rates, since the types of injuries associated with marathon are unlikely to occur if one does not run. The 10 km race was used as the reference category because it is the shortest of the three distances, typically involving lower physiological stress and shorter exposure to heat. This makes it a logical baseline to evaluate whether longer distances (half marathon, marathon) exhibit increased vulnerability to heat stress. Therefore, the shortest and expectedly least intense of the three runs was used as a baseline for calculation of IRRs for the other two runs. Additionally, the 10 km event had the largest number of participants, providing a stable reference group with sufficient events for comparison.

Owing to the changes in the way casualty information was collected, the number of seriously injured participants for each marathon being the constant item of information collected for each of the four years, the severe casualty rate was compared to the changes in the heat index over the years. Since complete information on injured participants was only collected for the 2016 marathon, the size of the less severe casualty pool was looked at to deduce the range of injuries sustained and in calculating the IRR for the half and full marathons.

## 5. Results

The marathon was held in or around the first week of December in all the four years covered. Weather conditions during the events for those four years are as given in Table 3. Most significantly, temperatures and heat indices were lowest in 2013 at the Caution range (based on the Classification of Heat Index) [12]. The Caution range reflects heat index at 27–32 °C during which fatigue is possible with prolonged exposure and/or physical activity. For the years from 2014 to 2016, the heat index was in the Extreme Caution range in the 32–39 °C range during which the risks of heat injuries are expected to be greater with prolonged exposure and/or physical activity.

The number of race participants from 2013 to 2016 for the different runs are as shown in Table 1. Table 4 shows that the number of injured participants documented ranged from 407 in 2013 to 3321 in 2016. In total, there were 5942 injured participants documented. There were more males than females injured participants every year. Their mean age was 36.9 ± 10.7 years. More than 96% of them were discharged after treatment.

There were more injured participants from the full marathon than the half marathon and the 10 km race in most of the years.

Casualty frequency by severity is given in Table 5. The numbers of severe casualties remained between 54 and 168 per marathon, with a PPR that peaked in 2014. Almost all the severely injured participants were managed at the finishing point tent. Participants with severe injuries were those with myocardial infarction, cardiac arrest, asthma, moderate to severe heat exhaustion or heat stroke, seizures, and significant head injury. There were two instances of cardiac arrest in the four years, of whom one survived. Therefore, the incidence of sudden cardiac arrest in the three race categories studied was 1.03 per 100,000 participants.

The 2016 data would better reflect the impact of the marathon on the overall proportion of light injuries for which the PPR was 66.13 per 1000 participants. The data suggests that medical tents managed only about 10.4% of all lightly injured participants, the open spaces near the finishing point about 15.7% to 23.4% of such participants and the remaining approximately 70% were managed along the marathon route in between the various medical tents. The majority of participants with light injuries presented with mild heat exhaustion, cramps, sprains, abrasions, bruises, contusions, and lacerations.

The very few moderately injured participants presented with stable injuries, such as contusions of the back, chest, face, and minor fractures.

The negative binomial regression showed a significant impact of heat index on incidence of severely injured participants (Table 6). When the heat index increased by one unit, the PPR for severely injured participants increased by a factor of 1.22 (a 22% higher casualty rate). Compared with 10 km participants, while holding heat index constant, half marathon participants experienced 1.58 times greater likelihood of injuries. Full marathon participants, on the other hand, had an IRR 3.87 times greater than for the 10 km participants.

The combined impact of both would be that a full marathon conducted in 35 °C heat index had 8.6 times more injuries than a 10 km run in 29 °C heat index (1.22^6^ × 3.87). Figure 1 below shows the predicted incidence rates of medical presentations (from the negative binomial model) across the observed heat index range, stratified by race distance. This approach preserves the continuous nature of the heat index variable while allowing visual comparison across race categories.

## 6. Discussion

The objective of this study was to determine the types of injuries occurring in marathon events being conducted in a tropical environment situated very close to the equator and with high temperatures and high relative humidity, so as to provide a basis for medical planning for future events in similar environments. This study showed an increasing trend for severe injuries with increasing heat index, a large incidence of heat exhaustion and muscular injuries in those with less severe injuries and an increasing transfer to hospital trend with increasing heat index. This is probably the first time that a study has looked at the likelihood of increasing heat index resulting in more severely injured participants and a significant number of heat-related injuries during high-intensity endurance events such as marathon runs, especially in a hot and humid environment.

Previous studies have shown that a significant number of participants experience injuries or illnesses during marathon training and races with an overall injury/illness rate of 20.0% [13]. This figure is documented from events occurring in cooler and less humid environments. Our PPR was higher than that of the 2012–2015 Chicago Marathon (31 per 1000), and the 45 per 1000 of the Vancouver Marathon [14,15]. Our overall PPR of 69.7 per 1000 participants, most of whom were recreational participants, appears high, even for tropical environments. For recreational participants who have suffered a running-related injury in the previous 12 months, the risk of re-injury would be expected to be higher [8]. However, no information on the history of previous injuries in our cohort of recreational marathon participants who participated in the Singapore Marathon was available. It is worth considering whether the higher heat index encountered when running in Singapore would have likely contributed to the higher injury rate.

A systematic review of sudden cardiac deaths during marathons revealed a rate of 0.6 to 1.9 per 100,000 runners [16]. This was based on events conducted in temperate environments. A 15-year review of cardiac arrests in Japan during marathons showed a sudden cardiac arrest rate of 2.18 per 100,000 participants [17]. This is not dissimilar to the experience from England (2.17 per 100,000 participants), and from the U.S. (0.54 per 100,000 participants); all of which, again, occurred in temperate climates [18,19,20,21]. Our sudden cardiac arrest incidence of 1.03 per 100,000 was well within that range and comparable to the incidence of 1.3 reported for the Hong Kong Standard Chartered Marathon, though the participant population and weather conditions in Hong Kong may have been very different [22]. The incidence is also similar to the rate of 1.01 reported in a large registry study [23]. Therefore, the largest contributor to the relative high injury rate may be injuries suffered by lightly injured participants.

Our study revealed an overall transport to hospital rate of 0.30 to 0.68 per 1000 participants. Transport to hospital would, almost always, be for severely injured participants. The higher rates were in the years when the heat index was higher. A study of 10 km run participants in Massachusetts, USA, showed a hospital transport rate of 0.2 per 1000 finishers [24]. In a four-year study of casualties sustained during the Baltimore marathons, 4.0% of patients who reported to the medical aid stations or 0.43 per thousand runners required transportation to a hospital [25]. The transport to hospital rates for the Chicago and Vancouver marathons were 0.48 and 0.09–0.58 per 1000, respectively [14,15]. Transport to hospital rates depend on the severity of injuries suffered by the marathon participants. It is to be expected that the higher likelihood of heat-related injuries in tropical Singapore with its high-heat index would contribute to the higher transport rates observed.

Temperatures, relative humidity and, therefore, heat indices in most marathons have been relatively low, compared to the levels in Singapore [26]. An increase in heat index from 2013 to 2016 correlated with an increase in the number of severely injured participants. The negative binomial model helped in estimating the relationship between changes in the heat index and the number of severely injured participants, providing an IRR that quantified the impact. In environments such as Singapore with an unusually high-heat index, it becomes all the more important for race organisers to address the ambient high-heat index and actively plan for measures to reduce the impact of heat on marathon participants. Future running events in any jurisdiction should consider the heat index for purposes of medical operations planning, as the risk of serious heat-related injury is significantly increased with increasing heat index. With careful planning and active measures to mitigate the effects of high heat on the participants, there should not be a need for the large numbers of heat injuries that are usually encountered during marathon events.

The heat index is a measure of how hot it feels to the human body when relative humidity is combined with the air temperature. It is particularly relevant to marathon running because it reflects the body’s ability to cool itself through sweating. During runs, the body produces heat. To cool down, sweating and evaporation of that sweat from skin are primary cooling mechanisms. High humidity reduces the rate of evaporation, making the body feel hotter than the actual air temperature. This is why a day with a high temperature and high humidity feels much hotter than a day with the same temperature but low humidity. The heat index combines both temperature and humidity to give a more accurate representation of how challenging it is for the body to cool down. A high-heat index means the body is under greater stress to maintain a safe core temperature. During a marathon, participants generate a lot of heat for prolonged periods. If the heat index is high, their bodies struggle to remove this heat. This significantly increases the risk of heat-related illnesses, which can range from heat cramps and heat exhaustion to heatstroke. Marathon organisers and participants must pay close attention to the heat index to make informed decisions about race safety, individual pacing and hydration strategies.

There are many strategies to minimise the risks of heat injuries for marathon participants. Race organisers may consider adjusting race timings, if possible, so as to schedule the race either in the early morning (which has already been implemented for the Singapore Marathon) or late afternoon to avoid the hottest part of the day. They would also have to provide adequate hydration for the runners with plentiful water and electrolyte drinks along the course, especially in high heat-index situations, offer multiple cooling stations with misting fans along the route, cold-water immersion, or ice to help runners cool down at the medical tents and finishing points [27]. The London and Boston marathons have one water point at every mile (1.6 km) between the 3rd and 25th mile, even though the heat indices in these locations are much less than in tropical environments [28,29]. Event planners in the tropics would need to consider emulating this to address the higher heat-injury risks at their locations. Participants also need to be educated about the risks of heat illness, symptoms to watch for, and how to prevent them. Medical support organisers will need to have trained medical personnel on-site to recognise and treat heat injuries promptly. In extreme heat and humidity conditions, race organisers may need to consider changing the venue of the race or even postponing the event.

Participants, on the other hand, need to acclimatise to the heat and gradually increase their exposure to warm conditions during training. They must hydrate properly by drinking plenty of fluids before, during, and after the race, and consider electrolyte drinks. During the run, they would need to pace themselves, avoiding starting too fast and adjusting their pace according to weather conditions and how they feel [27]. They should also wear appropriate clothing, choosing lightweight, breathable, light-coloured clothing that wicks away sweat and consider using cooling aids, such as ice vests, cooling towels, or spraying themselves with water. In addition, they should learn to identify the signs of heat illness (cramps, exhaustion, stroke) and stop running if these occur.

Most of the participants presented along the race route and not at the medical tents. The tent at the finish points usually sees the largest number of injured participants, whether the race is held in Singapore or the USA [8,30]. The commonest presenting complaints in the four years of the race were, in addition to obvious heat injuries, cramps, aches, and musculoskeletal injuries. This pattern was similar in other marathon studies. In a review by Ellapen et al., the predisposing factors for musculoskeletal injuries include poor training habits, incorrect shoes, and high weekly mileage [31]. Marathon events, owing to their high endurance nature, place significant stress on the body, leading to a variety of potential injuries. Some of the more common injuries include common overuse injuries, such as patellofemoral pain syndrome, iliotibial band syndrome, shin splints, Achilles tendinopathy, plantar fasciitis, and stress fractures. In addition, muscle strains, blisters, chafing, and ankle sprain are also seen frequently [32,33]. Factors contributing to such injuries include overtraining, improper footwear, poor running form, inadequate warm-up or cool-down, and sometimes, muscle weakness or imbalance and pre-existing conditions. Understanding these common injuries and their causes can help participants take preventive measures, such as proper training and appropriate footwear, to ensure a safe and successful marathon experience. Mandatory pre-race physical activity screening questionnaires may also need to be considered. The Singapore race website had a resource pack that included a structured 16-week training programme for beginner, intermediate, and advanced participants, and a nutrition guide. Running workshops are also conducted prior to race day. All first-aiders for similar running events undergo some form of training to manage musculoskeletal injuries as these form the bulk of the presentations. There is some evidence to suggest that online educational prevention programmes have no effect on the number of running-related injuries in recreational race participants [34].

### Limitations

There was some missing data in the casualty records, for example, disposition or presenting complaint. We tried to clarify the missing fields based on the records of treatment rendered, and from the annual final reports by the medical operations team to the race organisers after the races. It is also possible that some participants received help from medical staff but did not have their encounters captured in the records. These encounters were probably for mild conditions not requiring professional medical attention. Individual-level covariates (e.g., age, sex, medication use) were unavailable for all participants and thus could not be included, which may lead to residual confounding.

Finally, the figures provided by the race organisers for the 2016 marathon were rounded off by them before being sent to us resulting in less accurate numbers for that year compared to previous years.

## 7. Conclusions

Notwithstanding the health benefits of running, there is a significant casualty load at major racing events, with serious life-threatening injuries, sometimes. However, there is a paucity of published data regarding mass running events in tropical environments. For a popular race like a marathon, participant safety and well-being should be a top concern for race organisers. It is critical to study the gaps in the medical response plan yearly and make improvements to reduce morbidity, especially with regard to the heat factor. Evacuation plans must be clearly communicated and practised. We hope that this study will enable organisers and medical personnel to refine the logistics and manpower staffing for each sector of the race. Race participants must also be educated on proper race preparation and exercise responsibility.

## Figures and Tables

**Figure 1 healthcare-13-02249-f001:**
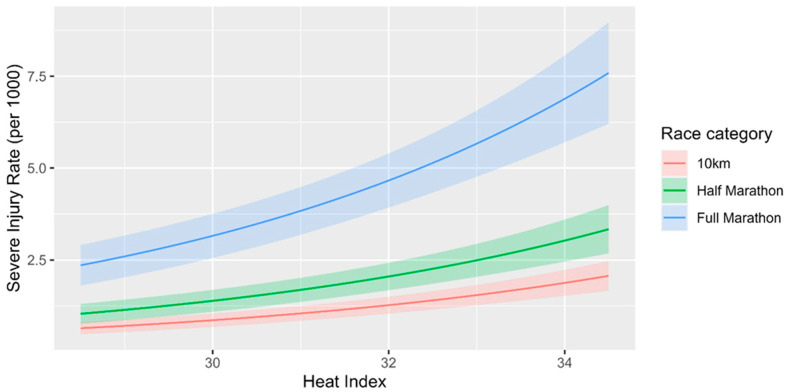
Predicted incidence rate of severe injuries vs. heat index by race distance.

**Table 1 healthcare-13-02249-t001:** Participant numbers and flag off times from 2013 to 2016.

	2013 Flag-Off Time	2013 Participant Numbers	2014 Flag-Off Time	2014 Participant Numbers	2015 Flag-Off Time	2015 Participant Numbers	2016 Flag-Off Time	2016 Participant Numbers
**Full Marathon**	0500 h	17,900	0500 h	14,322	0500 h	13,350	0430 h	15,000
**Half Marathon**	0630 h	12,000	0630 h	13,000	0630 h	12,005	0430 h	13,000
**10 km Run**	0715 h	20,000	0715 h	21,500	0715 h	21,871	0645 h	20,000
**Total**	-	49,900	-	48,822	-	47,226	-	48,000

**Table 2 healthcare-13-02249-t002:** Casualty codes used for the Singapore Marathon.

Severe Casualties	Moderate Casualties	Light Casualties
S0 (Acute Myocardial Infarction)	M0 (Abdominal Injury)	L0 (Abrasions, Bruise, Contusions)
S1 (Asthma)	M1 (Back Injury)	L1 (Cramps, Lower Limb)
S2 (Collapse)	M2 (Chest Injury)	L2 (Cramps, Upper Limb)
S3 (Gastrointestinal problem)	M3 (Facial Injury)	L3 (Cramps, stitches)
S4 (Heart Problem)	M4 (Fracture, Lower Limb)	L4 (Cuts/Lacerations)
S5 (Heat Exhaustion)	M5 (Fracture, Upper Limb)	L5 (Sprain, Ankle)
S6 (Heat Stroke)	M6 (Head injury, Minor)	L6 (Sprain, Back)
S7 (Hyperventilation)	M7 (Neck Injury)	L7 (Sprain, Hip)
S8 (Head Injury, Major)	M8 (Pelvic Injury)	L8 (Sprain, Knee)
S9 (Physical Exhaustion)		L9 (Sprain, Others)
S10 (Seizures/Fits)		

**Table 3 healthcare-13-02249-t003:** Temperature and relative humidity at marathon route.

	2013	2014	2015	2016	*p* Value
Temperature on Eastern part of Route (Range)	26 (25.4, 26.7)	27.9 (26.1, 30.8)	28.6 (27.2, 30.5)	28.1 (27.2, 29.4)	0.002
Relative Humidity Eastern part of Route (Range)	90.3 (85.6, 93.7)	77.5 (62.2, 88.5)	82 (72.0, 90.4)	86.3 (75.2, 94.7)	0.011
Heat Index (Eastern) (deg C)	28	31	34	33	-
Temperature at Western half of Route (Range)	26.3 (25.5, 27.6)	28.8 (26.2, 33.4)	29.1 (27.1, 32.2)	28.8 (26.5, 32.2)	0.034
Relative Humidity Western part of Route (Range)	89.2 (83.4, 92.3)	74.8 (56.1, 86.3)	78.7 (67.7, 88.8)	81.5 (61.0, 94.6)	0.056
Heat Index (Western) (deg C)	29	33	35	35	-

**Table 4 healthcare-13-02249-t004:** Overall injured participant and outcome numbers by year.

		Year				
Variable	Total	2013	2014	2015	2016	*p* Value
Number of Casualties	5942	407	1036	1178	3321	
Age (mean ± SD)	36.9 ± 10.7	36.1 ± 11.2	35.9 ± 10.7	37.1 ± 11.1	37.2 ± 10.6	0.001
Gender:						<0.001
Female, n (%)	1342 (22.6%)	70 (17.2%)	322 (31.1%)	284 (24.1%)	666 (20.1%)	
Male, n (%)	4600 (77.4%)	337 (82.8%)	714 (68.9%)	894 (75.9%)	2655 (79.9%)	
10 km Run, n (%)	726 (12.2%)	34 (8.4%)	195 (18.8%)	213 (18.1%)	284 (8.6%)	<0.001
Half Marathon, n (%)	1424 (24.0%)	42 (10.3%)	435 (42.0%)	255 (21.6%)	692 (20.8%)	
Full Marathon, n (%)	3792 (63.8%)	331 (81.3%)	406 (39.2%)	710 (60.3%)	2345 (70.6%)	
Outcome:						<0.001
Discharged, n (%)	5850 (98.5%)	392 (96.3%)	1018 (98.3%)	1146 (97.3%)	3294 (99.2%)	
Admitted, n (%)	92 (1.5%)	15 (3.7%)	18 (1.7%)	32 (2.7%)	27 (0.8%)	

**Table 5 healthcare-13-02249-t005:** Patient presentation rate and transport to hospital rate.

	2013	2014	2015	2016
Participant Numbers	49,900	48,822	47,226	48,000
Reported Casualty Numbers	407	1036	1178	3321
PPR (per 1000 participants)	8.16	21.22	24.94	69.19
Light casualties	353	865	1082	3174
Moderate casualties	0	3	3	4
Severe casualties	54	168	93	143
PPR * for light casualties	7.07	17.72	22.91	66.13
PPR * for severe casualties	1.08	3.44	1.97	2.98
TTH ^#^ numbers	15	18	32	27
TTH ^#^ (per 1000 participants)	0.30	0.37	0.68	0.56
TTH ^#^ (as % of severe casualties)	27.8%	10.7%	34.4%	29.7%

PPR * refers to patient presentation rate, ^#^ TTH refers to transport to hospital rate.

**Table 6 healthcare-13-02249-t006:** Negative binomial model: incidence rate ratios for heat index and type of competition.

	Incident Rate Ratio (IRR)	Incident Rate Ratio (IRR)	95% CI (Lower)	95% CI (Upper)	*p* Value
Heat Index *	1.22	1.22	1.19	1.25	<0.001
Half Marathon ^#^	1.58	1.58	1.42	1.76	<0.001
Full Marathon ^#^	3.87	3.87	3.50	4.28	<0.001

* The IRR reflects the increase in severe casualty rates for every unit rise in heat index from 2013 to 2016; ^#^ The IRRs for the half and full marathons are based on the presumptive value for the 10 km run as 1.00.

## Data Availability

Dataset available on request from the authors owing to confidentiality of personal information provided by the study participants, even though the data were anonymized.

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
