# Peer review of "Casualties During Marathon Events and Implications for Medical Support"

_healthcare, 2025, doi:10.3390/healthcare13172249_

Round 1

Reviewer 1 Report

Comments and Suggestions for Authors

The study’s chosen methods are generally appropriate but would benefit from clearer justification and detail in several areas:

  1. Data Collection and Case Definitions
    The retrospective collection of medical encounters from 2013–2016 provides a solid observational basis, but the manuscript should more explicitly justify why only “light” injuries were captured outside of the medical tent beginning in 2016. This change in case ascertainment introduces potential bias. A tabulated summary of which injury categories were recorded each year would help readers assess consistency and comparability across seasons.

  2. Injury Classification
    Classifying cases into “light,” “moderate,” and “severe” is reasonable; however, the criteria for each category are not fully described. The authors should specify the clinical thresholds or protocols used in the field (e.g., vital-sign cut-offs, need for evacuation) so that other researchers can reproduce the classification.

  3. Statistical Analysis
    Using incidence rate ratios (IRRs) to quantify the effect of heat index on different race distances (10 km, half-marathon, marathon) is appropriate. Yet, the choice of 10 km as the reference category requires justification: why was this distance deemed the baseline for comparison? Additionally, conducting a sensitivity analysis—perhaps by treating heat index as a continuous predictor or applying spline functions—would strengthen confidence that the results are robust to model specifications.

  4. Temporal Trends
    The methods describe pooling data across four years but do not account for secular changes in medical support or runner behavior. Incorporating year as a covariate or fitting a mixed-effects model with “year” as a random effect would control for unmeasured temporal confounders.

  5. Heat Index Measurement
    The daily heat index values were obtained from a single regional weather station. While practical, this approach assumes uniform environmental exposure for all participants. Discussing the potential for microclimate variation along the course—and, if possible, referencing nearby station data or course-specific measurements—would provide a more nuanced interpretation.

  6. Additional Comments on Results Presentation

    • Figures to Illustrate Key Findings

      • Consider adding a line chart showing the overall and severe PPR trends as a function of heat index across the four years.

      • Include a bar (or stacked-bar) chart depicting the distribution of medical presentations by race category (10 km, half-marathon, marathon) stratified by heat index bands.

    • Table Readability

      • Avoid merged cells that span multiple rows or columns; this will make the tables easier to parse.

      • Place any table footnote markers immediately following the variable label (e.g., “Temperature¹”) rather than before it, so that readers can quickly match notes to values.

In sum, the methodological framework is sound, but clarifying case definitions, justifying analytic choices, and addressing temporal and spatial variability would improve reproducibility and validity.

Comments on the Quality of English Language
  • Replace all instances of “casualty” and “runner” with a single, consistent term—suggest using “participant” throughout.

  • Fix minor typos such as “Tempertaure” → “Temperature” in Table 2.

Author Response

Reviewer 1 Comments and Responses

The study’s chosen methods are generally appropriate but would benefit from clearer justification and detail in several areas:

Comment 1. Data Collection and Case Definitions: The retrospective collection of medical encounters from 2013–2016 provides a solid observational basis, but the manuscript should more explicitly justify why only “light” injuries were captured outside of the medical tent beginning in 2016. This change in case ascertainment introduces potential bias. A tabulated summary of which injury categories were recorded each year would help readers assess consistency and comparability across seasons.

Response 1: Thank you for this enquiry. In 2013 only data about patients who were managed at the medical tents and those evacuated from site directly to hospitals was collected because of the expectation that most casualties would be brought to the medical tents for further care and subsequent disposition. This was extended to those seen in the open areas by physiotherapists at the large field near the finishing point for 2014 and 2015 because of the relatively large numbers of light casualties and resources expended to attend to them during the 2013 marathon, though data on these light casualties were not initially collected in 2013. In 2016, data of patients attended to by first-aiders all along the route, even if not referred to the medical tents or not evacuated directly to hospital, was also collected to obtain a more complete picture of injuries sustained and better appreciate the full scope of casualties sustained during the event. These reasons have been added to the manuscript.

Therefore, it wasn’t that only light injuries were captured outside the medical tent in 2016. All casualties, whether light, moderate or severe were attended to and information collected regardless of whether they occurred in the medical tents or outside the medical tents during the conduct of that marathon. Tables 4 and 5 provide a summary of the numbers and types of documented injuries by year.

Comment 2. Injury Classification: Classifying cases into “light,” “moderate,” and “severe” is reasonable; however, the criteria for each category are not fully described. The authors should specify the clinical thresholds or protocols used in the field (e.g., vital-sign cut-offs, need for evacuation) so that other researchers can reproduce the classification.   

Response 2: Thank you for the enquiry. The criteria for the field medical teams to classify the casualties into severe, moderate and light are as in the Table below

We have incorporated this table into the Methods section of the manuscript as Table 2.

Table 2: Casualty Codes used for the Singapore Marathon

Severe Casualties

Moderate Casualties

Light Casualties

S0 (Acute Myocardial Infarction)

M0 (Abdominal Injury)

L0 (Abrasions, Bruise, Contusions)

S1 (Asthma)

M1 (Back Injury)

L1 (Cramps, Lower Limb)

S2 (Collapse)

M2 (Chest Injury)

L2 (Cramps, Upper Limb)

S3 (Gastrointestinal problem)

M3 (Facial Injury)

L3 (Cramps, stitches)

S4 (Heart Problem )

M4 (Fracture, Lower Limb)

L4 (Cuts / Lacerations)

S5 (Heat Exhaustion)

M5 (Fracture, Upper Limb)

L5 (Sprain, Ankle)

S6 (Heat Stroke)

M6 (Head injury, Minor)

L6 (Sprain, Back)

S7 (Hyperventilation)

M7 (Neck Injury)

L7 (Sprain, Hip)

S8 (Head Injury, Major)

M8 (Pelvic Injury)

L8 (Sprain, Knee)

S9 (Physical Exhaustion)

L9 (Sprain, Others)

S10 (Seizures / Fits)

Comment 3. Statistical Analysis
a. Using incidence rate ratios (IRRs) to quantify the effect of heat index on different race distances (10 km, half-marathon, marathon) is appropriate. Yet, the choice of 10 km as the reference category requires justification: why was this distance deemed the baseline for comparison?

Response 3a: We thank the reviewer for this thoughtful and constructive comment. The IRR is a way to quantify how much more or less likely an event is to occur in one group compared to another, based on their respective incidence rates. In the case of a marathon event, using a non-run event was considered inappropriate for calculation of injury incidence rates, since the types of injuries associated with marathon are unlikely to occur if one does not run. The 10 km race was used as the reference category because it is the shortest of the three distances, typically involving lower physiological stress and shorter exposure to heat. This makes it a logical baseline to evaluate whether longer distances (half-marathon, marathon) exhibit increased vulnerability to heat stress. Therefore, the shortest and expectedly least intense of the three runs was used as a baseline for calculation of IRRs for the other two runs. Additionally, the 10 km event had the largest number of participants, providing a stable reference group with sufficient events for comparison. We have added this explanation to the section on Data Analysis.

  1. Additionally, conducting a sensitivity analysis—perhaps by treating heat index as a continuous predictor or applying spline functions—would strengthen confidence that the results are robust to model specifications.

Response 3b: Thank you for the suggestion. The negative binomial model that we used is one method of conducting sensitivity analysis. We had included Heat Index as a continuous predictor in our model. We did explore modelling heat index using restricted cubic splines. However, the heat index variable in our dataset had only four distinct values. The limited range and numbers of unique values were insufficient to support flexible non-linear modelling with spline functions, which typically require more data points to estimate smooth and stable curves.

Comment 4. Temporal Trends: The methods describe pooling data across four years but do not account for secular changes in medical support or runner behavior. Incorporating year as a covariate or fitting a mixed-effects model with “year” as a random effect would control for unmeasured temporal confounders.

Response 4: We thank the reviewer for highlighting the potential influence of secular trends. We had only one observation of incidence rate of injuries per race distance per year, for a total of four years. This limited structure does not support the use of a mixed-effects model with “year” as a random effect, as there are insufficient within-year replicates owing to lack of data in participants who had no injury. The changes in medical support over the four years were also minimal and would not be likely to significantly affect the incidence of injuries sustained.

Comment 5. Heat Index Measurement: The daily heat index values were obtained from a single regional weather station. While practical, this approach assumes uniform environmental exposure for all participants. Discussing the potential for microclimate variation along the course—and, if possible, referencing nearby station data or course-specific measurements—would provide a more nuanced interpretation.

Response 5: We thank the reviewer for raising this important point. Heat index data were available from two weather stations located on the race routes. Most of the runs were conducted in the western sector. Only a portion of the full marathon run (about 10 km of it) was conducted in the eastern sector.  From table 2, we can see the differences in temperature and humidity readings between these two stations were minimal: difference in temperature was less than 1 °C and humidity difference was between 1 to 5. Therefore, we used the mean of the heat index values from both locations to better represent the environmental exposure experienced by participants when calculating IRR values. We acknowledge that microclimatic variation may exist along the race routes owing to factors such as urban structures, shading, or elevation changes. However, given the relatively close proximity and overlapping of the routes (10 km to marathon) and the close similarity in weather data from both stations, we believe the averaged heat index provides a reasonable approximation of ambient conditions during the races for determination of the IRR.

Comment 6. Additional Comments on Results Presentation

  1. Figures to Illustrate Key Findings
      • Consider adding a line chart showing the overall and severe PPR trends as a function of heat index across the four years.
      • Include a bar (or stacked-bar) chart depicting the distribution of medical presentations by race category (10 km, half-marathon, marathon) stratified by heat index bands.

Response 6a: We agree with the reviewer that visualization of medical presentation counts by heat index would be valuable in principle. However, in our dataset, the heat index varied minimally across the four years. As such, a plot with raw data may only provide limited information on the association. Instead, we have generated an alternative figure showing the predicted incidence rates of medical presentations (from the negative binomial model) across the observed heat index range, stratified by race distance. This approach preserves the continuous nature of the heat index variable while allowing visual comparison across race categories.

Figure 1: Predicted Incidence Rate of Severe Injuries vs Heat Index by Race Distance

We have included this figure in the revised manuscript.

    1. Table Readability
      • Avoid merged cells that span multiple rows or columns; this will make the tables easier to parse.
      • Place any table footnote markers immediately following the variable label (e.g., “Temperature¹”) rather than before it, so that readers can quickly match notes to values.

Response 6b; thank you for the advice. We have made the necessary changes in the Tables to reflect this.

In sum, the methodological framework is sound, but clarifying case definitions, justifying analytic choices, and addressing temporal and spatial variability would improve reproducibility and validity.

Comment 7. Comments on the Quality of English Language

  • Replace all instances of “casualty” and “runner” with a single, consistent term—suggest using “participant” throughout.
  • Fix minor typos such as “Tempertaure” → “Temperature” in Table 2.

Response 7: Thank you for the suggestions. The words casualty and runner have been replaced with the word “participant” where applicable. We have also corrected the spelling error in the Table.

Reviewer 2 Report

Comments and Suggestions for Authors

The study spans four years, providing a solid dataset from the Standard Chartered Singapore Marathon that analyses casualty rates and trends over time.

The novelty of this study lies in presenting how the specific geographical weather conditions influence the safety of long-distance professional and amateur runners. It can mark a benchmark for possible future research and comparative analysis with marathons taking place in similar tropical climates.

Applying the negative binomial regression enhances the reliability of the findings by quantifying the impact of heat index on casualty rates.

The term "heat index" is mentioned repeatedly, as an important variable in the analysis, but lacks a clear definition or explanation for readers who may not be familiar with it. From the Abstract, one can understand that this is the highest temperature of the racing day, while the heat index is calculated using a formula that takes into account both temperature and humidity. 

Detailed information on injury types and transport-to-hospital rates (THR) provides actionable knowledge for medical teams. Furthermore, injury risks across participants of different races: 10km, half marathon, full marathon, allow for tailored interventions.

If authors referred to the number of casualties differentiated by gender, I strongly recommend mentioning the percentage of men and women registered in all races subsumed into the Singapore Marathon. Thus, we will have an explanation related to the predominance of injuries among male participants.

Regarding the 10km race, which is almost a quarter of the length of the
42,195m marathons, I deduce that it took place only in the eastern part
of the island, where the temperature and heat index are lower. Is that true?
The clarity of statements eliminates ambiguity and assumptions, which may be true, partially true, or false.

The conclusions suggest measures to minimise heat-related injuries. Among those, there is a suggestion "organizers may consider adjusting race timings, if possible, to schedule the race in the early morning". Is there an earlier time in the morning than 4.30 (like in 2016) when the sun rises at 7.00? 

There is another measure difficult to implement in an event with 50,000 local and international, amators and professional athlets that involves a significant number of organisers, logistics, and important funds: "Organisers may also need to modify the race and, in extreme conditions, consider shortening the race, altering the course, or even cancelling the event". I suggest a nuance of this recommendation that could be applicable.                                                                    Do you consider it appropriate to add a graphic with the parallel evolution of medical casualties and the heat index over the four editions of the Singapore Marathon?

The findings are particularly relevant for marathons in tropical regions, but they also offer insights applicable to other environments and mass sporting events.

Author Response

Reviewer 2 Comments and Response

Comment 1. The study spans four years, providing a solid dataset from the Standard Chartered Singapore Marathon that analyses casualty rates and trends over time.

The novelty of this study lies in presenting how the specific geographical weather conditions influence the safety of long-distance professional and amateur runners. It can mark a benchmark for possible future research and comparative analysis with marathons taking place in similar tropical climates.

Applying the negative binomial regression enhances the reliability of the findings by quantifying the impact of heat index on casualty rates.

Response 1: Thank you.

Comment 2. The term "heat index" is mentioned repeatedly, as an important variable in the analysis, but lacks a clear definition or explanation for readers who may not be familiar with it. From the Abstract, one can understand that this is the highest temperature of the racing day, while the heat index is calculated using a formula that takes into account both temperature and humidity. 

Response 2: Thank you for informing us about this. We have added a paragraph in the Methods section that introduces the reader to what the heat index is. This is as follows: “The heat index, also known as the apparent temperature, is a measure of how hot it feels to the human body when relative humidity is combined with the actual air temperature. It essentially reflects the human body's perception of heat, accounting for the cooling effect of evaporation being reduced by higher humidity.”

Further, in the Discussion section, we had provided a fairly large paragraph on details oh heat index. We have modified this slightly and the revised version is as such “The heat index is a measure of how hot it feels to the human body when relative humidity is combined with the air temperature. It is particularly relevant to marathon running because it reflects the body's ability to cool itself through sweating. During runs, the body produces heat. To cool down, sweating and evaporation of that sweat from skin are primary cooling mechanisms. So, when the humidity is high, sweat doesn't evaporate as efficiently. This is why a day with a high temperature and high humidity feels much hotter than a day with the same temperature but low humidity. The heat index combines both temperature and humidity to give a more accurate representation of how challenging it is for the body to cool down. A high heat index means the body is under greater stress to maintain a safe core temperature. During a marathon, runners generate a lot of heat for prolonged periods. If the heat index is high, their bodies struggle to get rid of this heat. This significantly increases the risk of heat-related illnesses, which can range from heat cramps and heat exhaustion to heatstroke. Marathon organizers and runners must pay close attention to the heat index to make informed decisions about race safety, individual pacing and hydration strategies.”

Comment 3: Detailed information on injury types and transport-to-hospital rates (THR) provides actionable knowledge for medical teams. Furthermore, injury risks across participants of different races: 10km, half marathon, full marathon, allow for tailored interventions.

If authors referred to the number of casualties differentiated by gender, I strongly recommend mentioning the percentage of men and women registered in all races subsumed into the Singapore Marathon. Thus, we will have an explanation related to the predominance of injuries among male participants.

Response 3: Thank you for this suggestion. Unfortunately, we were not provide with the figures on percentage of men and women registered in all races subsumed into the Singapore Marathon by the race organisers. We are, therefore, unable to provide this information.

Comment 4: Regarding the 10km race, which is almost a quarter of the length of the 42,195m marathons, I deduce that it took place only in the eastern part of the island, where the temperature and heat index are lower. Is that true? The clarity of statements eliminates ambiguity and assumptions, which may be true, partially true, or false.

Response 4: Thank you for informing us of this. The 10 km marathon took place in the western part of the island, where temperatures and heat index were higher. Most of the races occurred in the western part. Only a relatively small segment (about 10 km) of the full marathon occurred in the eastern part oc the island. This has been clarified in the Methods section as follows: “Only a relatively small portion of the full marathon (about 10 km) occurred in the eastern part of the island. The other races were held in the western part of the whole area used by the marathon event.”

Comment 5: The conclusions suggest measures to minimise heat-related injuries. Among those, there is a suggestion "organizers may consider adjusting race timings, if possible, to schedule the race in the early morning". Is there an earlier time in the morning than 4.30 (like in 2016) when the sun rises at 7.00? 

Response 5: Thank you for asking this. The suggestion was made as a general measure and not specific to the Singapore Marathon at which the start time has already been brought forward to 4:30 in the morning. The revised sentence in the Discussion section now reads as follows: “Race organizers may consider adjusting race timings, if possible, so as to schedule the race either in the early morning (which has already been done for the Singapore marathons) or late afternoon to avoid the hottest part of the day.”

Comment 6: There is another measure difficult to implement in an event with 50,000 local and international, amators and professional athletes that involves a significant number of organisers, logistics, and important funds: "Organisers may also need to modify the race and, in extreme conditions, consider shortening the race, altering the course, or even cancelling the event". I suggest a nuance of this recommendation that could be applicable. 

Response 6: Thank you for this point. We have amended the statement to read as follows: “In extreme heat and humidity conditions, race organisers may need to consider changing the venue of the race or even postponing the event”

Comment 7: Do you consider it appropriate to add a graphic with the parallel evolution of medical casualties and the heat index over the four editions of the Singapore Marathon?

The findings are particularly relevant for marathons in tropical regions, but they also offer insights applicable to other environments and mass sporting events.

Response 7: Thank you for the suggestion. We have added a graphic at the end of the Results section to reflect this. The addition is as follows:

Figure 1 below shows the predicted incidence rates of medical presentations (from the negative binomial model) across the observed heat index range, stratified by race distance. This approach preserves the continuous nature of the heat index variable while allowing visual comparison across race categories.

Figure 1: Predicted Incidence Rate of Severe Injuries vs Heat Index by Race Distance

Round 2

Reviewer 1 Report

Comments and Suggestions for Authors

Thanks for your reply and for making the revisions to the article. I have reviewed the changes and your responses, and I am pleased with the updated version.

Author Response

Comment 1: Thanks for your reply and for making the revisions to the article. I have reviewed the changes and your responses, and I am pleased with the updated version.

Reply 1: We thank Reviewer 1 for the comment.

Reviewer 2 Report

Comments and Suggestions for Authors

Super work!

Congratulate the authors!

Author Response

Comment 1: Super work! Congratulate the authors!

Response 1: We thank Reviewer 2 for the comments.